# Efficient Convolutional Neural Networks for Semiconductor Wafer Bin Map Classification

**DOI:** 10.3390/s23041926

**Published:** 2023-02-08

**Authors:** Eunmi Shin, Chang D. Yoo

**Affiliations:** Korea Advanced Institute of Science and Technology, Daejeon 34141, Republic of Korea

**Keywords:** wafer map, defect pattern, pattern classification, light-weight convolutional neural networks

## Abstract

The results obtained in the wafer test process are expressed as a wafer map and contain important information indicating whether each chip on the wafer is functioning normally. The defect patterns shown on the wafer map provide information about the process and equipment in which the defect occurred, but automating pattern classification is difficult to apply to actual manufacturing sites unless processing speed and resource efficiency are supported. The purpose of this study was to classify these defect patterns with a small amount of resources and time. To this end, we explored an efficient convolutional neural network model that can incorporate three properties: (1) state-of-the-art performances, (2) less resource usage, and (3) faster processing time. In this study, we dealt with classifying nine types of frequently found defect patterns: center, donut, edge-location, edge-ring, location, random, scratch, near-full type, and None type using open dataset WM-811K. We compared classification performance, resource usage, and processing time using EfficientNetV2, ShuffleNetV2, MobileNetV2 and MobileNetV3, which are the smallest and latest light-weight convolutional neural network models. As a result, the MobileNetV3-based wafer map pattern classifier uses 7.5 times fewer parameters than ResNet, and the training speed is 7.2 times and the inference speed is 4.9 times faster, while the accuracy is 98% and the F1 score is 89.5%, achieving the same level. Therefore, it can be proved that it can be used as a wafer map classification model without high-performance hardware in an actual manufacturing system.

## 1. Introduction

### 1.1. Background

The semiconductor manufacturing process consists of the front-end process and the back-end process. In the front-end process, processes such as oxidation, photo, etching, deposition, and ion implantation are repeatedly performed on the surface of the wafer to make the wafer into a semiconductor. In the back-end process, a wafer test process is performed to check whether all processes have been performed properly. Then, the packaging process of cutting and assembling wafers into chips and final quality inspections are performed. The wafer test process is important because it provides information to determine if a problem has occurred in the front-end process and whether each wafer is operating normally.

Through the wafer test process, the number of normal chips that can be used on one wafer is counted, and the ratio of the number of normal chips to the total number of chips is defined as the yield. The yield calculated here is used as a key indicator of semiconductor productivity, so it is managed with great importance. During the wafer test process, the defects of each chip are classified into several categories and stored in the form of letters. In order to check this information at a glance, a wafer bin map is made by distinguishing defective chips from normal chips and expressing them in different colors. Figure 1 is an example of a common defect pattern frequently found on wafers. It is a wafer map image for each defect pattern type obtained from the open dataset WM-811K used in this study. Clustered defect patterns, such as center, edge-loc, and loc, are commonly found when particles, which mean foreign substances, such as dust, are generated during the process. Circular patterns such as donuts often occur when the uniformity is low due to an improperly performed etching process that cuts off the rest of the circuit pattern. Types such as random and near-full could be wrong from the initial process. When a defect pattern like this occurs, it indicates a problem with a particular process or equipment. So it is filtered out through visual inspection and used for defect analysis. Even in the front-end process, there are several visual inspection processes, but they are only conducted on some sample wafers, whereas wafer tests are conducted on all wafers and are more important because they are directly related to yield. Therefore, in this study, we would like to focus on the problems related to the wafer bin map among visual inspections.

The entire semiconductor process generally takes more than 3 months to complete, and as the circuit becomes more refined, the process of the front-end process becomes more complex and more difficult, requiring more than 2 months. As the process becomes longer, the cost of production increases. To reduce this, the inspection process is minimized in the front-end process and a wafer test is performed after the front-end process is finished. The wafer test process is a process closely related to productivity because it includes the process of checking whether the wafer is good or not and repairing it if possible. Yields are usually calculated automatically, but still visual inspection of wafer patterns is performed manually by a human. However, as production volume continues to increase and factory automation accelerates, various attempts are being made to automatically classify wafer map patterns in order to reduce the manpower and time required for this task.

Pattern classification studies using well-known CNN models have already been performed, and a recent study showed that the classification accuracy of the model is over 95%. However, in most cases, in order to apply a classification model to a real system, new hardware investment is required because the time and hardware resources required for model training and inference are insufficient or may affect the existing system. Manufacturing companies tend to be reluctant to introduce systems that apply machine learning models due to the burden of investment and maintenance costs following the introduction of high-performance hardware. They are also concerned about an increase in total production time due to delayed response times of automated systems. Therefore, no matter how good the classification performance is, if excessive resources are used or the size of the model is too large, it cannot be used. In this study, we want to find a classification model that uses a minimum of resources and significantly reduces training and inference time while achieving good classification performance.

In addition, there are two main obstacles to the wafer map classification problem. The first is that it is difficult to obtain labels, which are information about defects or types of patterns. For general image pattern classification, there must be enough labels, but these labels do not exist in wafer maps gathered in actual manufacturing facilities. The second is that the dataset is imbalanced because the percentage of bad wafers in the data is much less than normal. Overfitting is a common problem when dealing with classification problems with imbalanced data. In this study, in order to minimize the side effects caused by the two problems mentioned above, the experiment was conducted by mitigating the data imbalance problem through data augmentation.

### 1.2. Contributions

In this study, we tried to find a way to improve the inefficient manual map pattern classification task, find an efficient deep learning classification model applicable to manufacturing, and compare it with the existing model to prove that good performance can be obtained even with a light-weight model. Light-weight models such as EfficientNetV2, ShuffleNetV2, and MobileNetV3, which have not been tried before, were applied to wafer map classification to confirm performance, and the applicability in the field was confirmed through performance comparison according to platforms as well as classification performance.

### 1.3. Structure of This Paper

The structure of this paper is as follows. Section 2 introduces related works on the topic. Section 3 deals with the method proposed in this study. Section 4 is about the experimental details and their results. Section 5 is the conclusion. Finally, Section 6 describes the limitations of this study and future work.

## 2. Related Works

### 2.1. Wafer Map Classification

A stacking ensemble model combining several manually extracted features with a convolutional neural network was proposed in [1]. Manual feature extraction involves extracting geometric, radon-based, and density-based features, and classifying the extracted features through machine learning models such as support vector machines, logistic regression, and feed-forward neural networks. Combining the two methods of classifying patterns through manual extraction and convolutional neural networks has been shown to improve classification performance compared to classification using each model.

To improve wafer map classification performance, [2] first extracted image features with a convolutional neural network model without preprocessing, and then applied pattern classification by combining error correction output code (ECOC) and support vector machine (SVM). Through experiments, it was demonstrated that the combined method of ECOC and SVM is effective in improving classification performance without preprocessing. Another study [3] considered that, although deep convolutional neural networks have made great strides in classifying wafer map patterns, they are difficult to apply due to the lack of real labeled data. They proposed a model pre-trained on unlabeled data via self-supervised learning and fine-tuned with labeled data.

In [4], in order to solve the problem caused by the imbalanced data distribution of the wafer map, good samples of defect pattern data are replicated or transformed in various ways to increase little defect pattern data. Then they showed that classification performance improves using a convolutional neural network model. Similarly, in [5], after making the number of data per class equal using data augmentation, patterns were classified with a model of a relatively simple 8-layer convolutional neural network structure, and performance was compared with other models.

Nakazawa and Kulkarni generated synthetic wafer maps with defect patterns via a deep convolutional encoder-decoder in [6]. The model used in this study succeeded in detecting a new type of defect pattern by learning eight basic defect patterns. Wang et al. proposed a model that can classify two or more mixed type defect patterns in [7].

In [8], Yoon and Kang proposed a semi-automated system that measures the uncertainty of wafer defect pattern classification results through reliability, entropy, and Gini coefficient. If the uncertainty is below the threshold, the system automatically classifies the defect pattern, and if the uncertainty is high, the system rejects the classification and allows the engineer to make a decision.

Tsai and Lee performed pattern classification through light-weight deep convolutional neural network models MobileNetV1 and MobileNetV2 in [9]. After augmenting the data with a convolutional encoder-decoder, a light-weight deep convolutional model has been shown to reduce model parameters and computational load and improve accuracy.

Chen et al. performed wafer map classification on the WM-811K dataset by combining a dual-channel CNN and an ECOC-SVM classifier in [10]. Zheng et al. performed wafer map classification by comparing ML-based and DL-based models, demonstrating that DL-based models achieve better performance [11]. In [12], high accuracy was achieved by image augmentation with G2LGAN followed by wafer map classification with MobileNetV2 classifier. In [13], Cha and Jeong combined improved U-Net with residual attention block to perform classification on mixed-type wafer maps.

Yu et al. demonstrated a good performance as a result of classifying the WM-811K dataset with only a very small number of parameters and FLOPs through PeleeNet-based WM-PeleeNet [14]. Doss et al. proposed a model that performed transfer learning over a pretrained ShuffleNetV2 model in [15].

Table 1 summarizes the accuracies and F1 scores presented in the related works. In the same way as in this study, the experimental results using the WM-811K dataset are summarized.

### 2.2. Light-Weight Convolutional Neural Networks

Iandola et al. proposed SqueezeNet [16], a small convolutional neural network model with AlexNet-like accuracy, 50× fewer parameters, and a model size of less than 0.5 MB on the ImageNet dataset. They designed the model through three strategies. The first strategy is to reduce the number of parameters by replacing the 3 × 3 filter with the 1 × 1 filter, and the second strategy is to reduce the number of parameters by reducing the number of input channels of the 3 × 3 filter. A third strategy is to maximize accuracy by using late downsampling in the network so that the convolutional layer has a large activation map. They introduced building blocks called fire modules that implemented strategies 1 and 2 and, combined with strategy 3, they designed SqueezeNet, a model that uses far fewer resources while retaining AlexNet-level accuracy.

Howard et al. proposed MobileNet, an efficient model based on depthwise separable convolution for mobile and embedded applications [17]. A depthwise separable convolution consists of a depthwise convolution and a pointwise convolution (1 × 1 convolution). In MobileNet, we reduce the computational cost by applying a single filter to each input channel and then combining the outputs of the 1 × 1 convolution and the depthwise convolution.

Zhang et al. proposed a ShuffleNet with a computationally efficient CNN structure optimized for mobile devices [18]. They used pointwise group convolution and channel shuffle to minimize computational cost while maintaining accuracy. In [18], a channel sparse connection method such as group convolution was applied. Group convolution can reduce computational cost because it performs operations between corresponding groups, but when multiple group convolutions are accumulated, it is difficult to share features because information flows only between specific channels. Therefore, a method called channel shuffle was used to connect channels between different groups to create a more robust structure.

Tan and Le introduced a new complex coefficient to show that models with better performance can be found by balancing the scale of network depth, width, and resolution. In order to improve the performance of convolutional neural networks, it is necessary to expand the model structure. The expansion method can be implemented by applying depth direction expansion, width direction expansion, input resolution expansion, or all three at the same time. However, as the size of the model scales up, the amount of computation also increases. Therefore, as a way to efficiently expand the model, they studied how to find the optimal combination of depth, width, and resolution, and also proposed EfficientNet [19] as a basic network.

As a follow-up study, MobileNetV2 [20] improved resource efficiency compared to MobileNetV1 by designing a structure using linear bottlenecks and inverted residuals. MobileNetV3 [21] uses network search to find the optimal network structure. The NetAdapt algorithm is used in the layer search method, which is a block-by-block search method called platform-aware neural network structure search. Ma et al. proposed ShuffleNetV2 [22], which improved ShuffleNet by finding a structure suitable for the target platform by considering direct metrics such as speed. EfficientNetV2 [23] is a model that points out three factors that slow down the learning speed and introduces a method to remove them in order to speed up the learning.

## 3. Method

The flowchart of wafer map classification proposed in this study is shown in Figure 2. The method is divided into two phases. In the training phase, we trained a light-weight CNN model by normalizing labeled wafer map images and mitigating imbalances through data augmentation. In the testing phase, after undergoing preprocessing that applies only normalization to the unlabeled wafer map image, the trained model was tested to predict the label.

### 3.1. Data Augmentation

Wafer bin maps contain important information related to wafer defects. In particular, defects that appear in a specific pattern become clues to the cause of major defects in the manufacturing process, so quality problems can be minimized through quick detection, cause analysis, and action. In general, since all wafers are tested in the wafer test process, all wafer bin maps are also required to be inspected. Currently, a person visually checks the wafer bin map one by one, and when there is a specific pattern, the wafer is filtered out or further analyzed. At this time, labeling is sometimes assigned according to somewhat subjective standards depending on the engineer, and above all, since the percentage of defects in the entire data is relatively small, it is difficult to obtain defective samples that can be labeled with a specific defect pattern. The WM-811K dataset used in this study also had the distribution of Table 2, and only a small number of samples exist for specific patterns.

This data imbalance problem can cause overfitting when training is performed through a classification model. Therefore, we additionally generated defect image data through data augmentation and used it for model training. Images in the dataset were normalized before use, and the training images were augmented by transformation. RandomRotation, RandomErasing, Resize, RandomCrop, GaussianBlur, RandomHorizontalFlip, and RandomVerticalFlip were applied as image transformation methods for augmentation. All of the aforementioned transformations were applied to each image, but all of them were randomly applied. As a result, images that have been transformed into various forms, such as in Figure 3, have been added.

### 3.2. Light-Weight Convolutional Neural Network Models

Pattern classification was performed using ShuffleNetV2, MobileNetV2, MobileNetV3, and EfficientNetV2 models, which are the state-of-the-art light-weight convolutional neural network models.

In the light-weight model, the main task was to secure the number of usable channels within limited resources, but channel shuffle used in ShuffleNet is a method to obtain the effect of increasing the number of channels without greatly increasing the amount of computation. Furthermore, a structure that additionally proposed various methods for more efficient structure design appeared in [22]. As a practical guideline, the first is to use balanced convolution to have the same channel width, the second is to consider the cost of group convolution, the third is to reduce the degree of network fragmentation that hinders parallel processing. The last thing is to reduce element-wise operations. Taking this into account, channel split operations were introduced in ShuffleNetV2 to keep the number of channels larger and wider. The basic unit using channel split can be represented as shown in Figure 4c,d. It is a downsampling unit for increasing the number of channels, which is performed without a channel split operation, and the number of channels doubles after passing this unit. ShuffleNetV2 is composed of (c) and (d) units in Figure 4, and the overall structure of the model used for wafer map classification is shown in Table 3. The size of the wafer map input image was given as 224 × 224, and the first convolution applied a filter with a size of 3 × 3 and a max pooling layer. Stages 2, 3, and 4 repeatedly used the basic unit and downsampling unit of ShuffleNetV2 to build layers and, in Conv5, 1 × 1 convolution was used. Right before the last fully connected layer, the feature map was reduced in the global average pooling layer and then classified into nine classes.

MobileNet even came out to V3, and we used V2 and V3 for performance comparison. The main differences between V1 and V2 and V3 are the linear bottleneck and the use of inverted residual blocks. The structure of MobileNetV2 used in this study is shown in Table 4. There are two types of block layers. The first block is an inverted residual block with a stride of 1, and the second block is a block with a stride of 2. In both blocks, the first layer is pointwise convolution and ReLU6, and the second layer is depthwise convolution. The third layer is again a pointwise convolution, but here there is no activation function. Each input and output has a constant t value called the expansion factor used to scale the output channel. A value between 5 and 10 is recommended for the constant t. As a result of the experiment, the best performance was shown at about 6, so t = 6 was also applied to this model.

MobileNetV3 uses the network search algorithm to find the optimal network structure while using the basic structure of version 2 as it is. The first of the major improvements was to replace the 1 × 1 convolution used in the last step of version 2 for expansion into a high-dimensional feature space with an average pooling layer, reducing the amount of computation without loss of accuracy. Secondly, instead of ReLU, a nonlinear activation function called swish was introduced. The swish function is defined as:(1)$$\begin{array}{c} swishx=x\cdot \sigma (x). \end{array}$$

Since the sigmoid function requires a considerable amount of calculation, a nonlinear function called swish was used instead of ReLU to reduce the amount of calculation. A hard version of the h-swish function with sigmoid changed to ReLU6 can be expressed as:(2)$$\begin{array}{c} h\text{-}swish\left[x\right]=x\frac{ReLU6(x+3)}{6}. \end{array}$$

Since these nonlinear functions have a computational cost saving effect in deep networks, h-swish was used as an activation function in the second half of the entire network. Third, using the squeeze-and-excite bottleneck structure used in the previous study [24], the number of channels in the extension layer was changed to be fixed to 1/4. In addition, MobileNetV3 is defined in two versions, a large model and a small model. Since the wafer map image works well enough even for a small model, the MobileNet V3-Small model with the structure shown in Table 5 was used in this study. In Table 5, SE represents a squeeze-and-excite block, NL represents nonlinearity, RE represents ReLU, HS represents hswish and NBN is a layer that does not use batch normalization.

EfficientNetV2 [23] studied the obstacles to model training. One of the contributing factors to long training times is the size of very large images, with larger image sizes resulting in higher memory consumption and longer training times. Another hurdle is that depthwise convolution works well for later layers of the network, but is slow for earlier layers. Since increasing the network size uniformly at every step is ineffective, EfficientNetV2 ameliorates this problem by fine-tuning the scaling rules, such as by incrementally increasing the image size and limiting the maximum image size. To improve the problem of depthwise convolution, 1 × 1 convolution and 3 × 3 depthwise convolution were converted into standard 3 × 3 convolution in the MBConv structure as shown on the left of Figure 5. A model replaced with Fused-MBConv with the structure shown on the right of the Figure 5 changed to a solution was proposed, and when applied to EfficientNet, the training speed was improved. We used EfficientNetV2-S with the same structure as in Table 6. Looking at the structure of Table 6, all steps do not use the same block, use Fused-MBConv blocks that do not use a depthwise convolution in the first half of the network, and MBConv blocks use a depthwise convolution in the second half of the network.

In the process of training the model, the cross entropy loss function was used to check whether the classification was successful. Cross entropy loss, commonly used for multi-class classification, is defined as:(3)$$\begin{array}{c} CELoss=-\sum_{i}^{C}t_{i}\log\left(f{\left(s\right)}_{i}\right). \end{array}$$$t_{i}$ means the correct answer, *s* means the score predicted by the model, and *C* means each class. The higher the uncertainty of the prediction result, the larger the value of the loss function, and the lower the uncertainty, the smaller the value of the loss function. During training, the model updated the parameters so that the value of the loss function became smaller.

## 4. Experiments

### 4.1. Dataset

For training, we used WM-811K, a data set consisting of 811,457 wafers collected from real fabrication. The WM-811K dataset consists of 172,950 labeled data and 639,507 unlabeled data, with 632 images ranging in size from 6 × 21 to 300 × 202. Label data were defined as a total of nine types, including eight types with different defect pattern types and the None type classified as normal wafers. In this study, 172,950 labeled data were converted into a 3-channel image with a size of 224 × 224 by dividing it into a training dataset of 80% and an evaluation dataset of 20%. In addition, as shown in Table 7, the number of images was increased to 10,000 for each class through data augmentation for the eight defect pattern training data points. The experimental method used the K-fold cross-validation method to measure generalized performance. Excluding the testing dataset, which is 20% of the total data, the training dataset was divided into four folds and separated into training and validation datasets. That is, the ratio of training:validation:testing data was set to 6:2:2. Three of the four folds were divided for training and one for validation, each model was trained and validated, and the accuracy and F1 score were measured. The validation fold was changed, the rest of the folds were repeated, the performance results were averaged for each fold and the standard deviation was obtained. In the case of the testing dataset, data distribution close to the actual data was maintained without data augmentation, and accuracy, precision, recall, and F1 score were measured.

### 4.2. Settings

The experimental environment was implemented using Python 3.7 and Pytorch, and the experiment was conducted with GPU on Quadro RTX 8000. The size of the input image for model training was unified into three channels of 224 × 224, the batch size was 128, the optimizer was Adam, and the learning rate was 1 × 10^−4^. In additional experiments, training and inference times were measured using GPU and CPU for each model. The CPU environment was tested on Intel(R) Core i7-9700F.

### 4.3. Evaluation Metric

Accuracy, precision, recall and F1 score were used to compare the performance of the models. The prediction result of the classifier and the confusion matrix of the actual value can be expressed as Table 8, and each index can be obtained as Equations (Equation 4)–(Equation 7).
(4)$$\begin{array}{c} Accuracy=\frac{TP+TN}{TP+TN+FP+FN} \end{array}$$
(5)$$\begin{array}{c} Precision=\frac{TP}{TP+FP} \end{array}$$
(6)$$\begin{array}{c} Recall=\frac{TP}{TP+FN} \end{array}$$
(7)$$\begin{array}{c} F1Score=\frac{2\times Precision\times Recall}{Precision+Recall}. \end{array}$$

We also compared how efficiently resources were being used by measuring the size of parameters, memory usage, MAdds, FLOPs, training time, and inference time. MAdds is the number of multiplication and addition operations and FLOPs is the number of floating point operations.

### 4.4. Experiment Results

As a result of the experiment, the accuracy and F1 scores of the training and validation datasets for each model are shown in Table 9. The accuracy, precision, recall, and F1 score of the testing dataset are shown in Table 10. The models used were ResNet18, which is well known as an image classification model, and state-of-the-art light-weight models such as EfficientNet V2-S, ShuffleNetV2, ShuffleNetV2 0.5×, MobileNetV2, and MobileNetV3-Small. CNN-WDI [5], at the end, was used in [5]. Compared to ResNet18, all five light-weight models used in the experiment in this study showed a slightly lower classification performance on training and validation datasets, but MobileNetV3 had the same F1 score as ResNet18 for the testing dataset. Compared to CNN-WDI [5], MobileNetV3 had higher training and validation F1 scores, and the same testing F1 scores, indicating a very good performance even though it is a light-weight model.

Looking at the experimental results of all models in Table 10, the F1 score was lower than the accuracy, because the evaluation dataset had an imbalanced distribution. In other words, since most of the data were of the None type with a lot of data, even if the model randomly classifies them as the None type, this is highly likely to be the correct answer with a high probability. Therefore, it is unreasonable to evaluate the performance of the model only on the basis of accuracy in this study. In fact, when the classification result is expressed as a confusion matrix for each class, it can be seen that data are concentrated in a specific class as shown in Figure 6.

Figure 7 shows the confusion matrix representing the classification results for each class as a normalized value between 0 and 1 to compare how well the models for each class classified. As can be seen from this normalized confusion matrix, some classes showed results close to 1, but some classes showed relatively low classification accuracy. Since these differences between classes cannot be sufficiently reflected in accuracy, the F1 score for each class was set as a performance evaluation index to reduce distortion of results due to imbalanced data distribution. Therefore, the final performance was evaluated based on the Macro F1 score, which gives equal weight to each class.

Table 11 shows the number of parameters, memory size, number of multiplication and addition operations, and number of floating point operations for each model. Compared to ResNet18, the number of parameters decreased by 8.6 times for ShuffleNetV2, 32 times for ShuffleNetV2 0.5×, 5.1 times and 7.5 times for MobileNetV2 and V3, respectively, and increased by 1.8 times for EfficientNetV2. In the case of memory, ShuffleNetV2 reduced by 1.2 times, ShuffleNetV2 0.5× by 2.5 times, and MobileNetV3 by 1.4 times. However, EfficientNetV2 and MobileNetV2 increased by 6.3 times and 3.4 times, respectively. The number of arithmetic operations and floating-point operations in ShuffleNetV2 is reduced by about 12 times compared to ResNet18, 44 times in ShuffleNetV2 0.5×, 5.8 times in MobileNet in V2, and about 30 times in V3.

Table 12 is the result of measuring execution time for each model and indicates the number of images processed per second during the model training and inference phases, based on experiments using GPU and CPU on ResNet18 and light-weight models. In terms of training speed, ShuffleNetV2 showed a 6.9 times faster processing speed, MobileNetV2 was 5.4 times faster, and MobileNetV3 was 7.2 times faster than ResNet18. It was very slow overall on the CPU, but still achieved the fastest results on MobileNetV3. In the case of inference speed, it was slow overall, but when compared relatively, MobileNetV3 was the fastest, followed by ShuffleNetV2, ResNet18, and MobileNetV2. EfficientNetV2 was excluded because the execution time on the CPU was excessively long, and ShuffleNetV2 0.5× was excluded due to its low F1 score. In real applications, models can be pre-trained on a GPU environment and then inferred on a CPU environment.

Figure 8 shows the number of parameters, number of floating-point operations, and F1 score to compare resource utilization and performance for each model. The smaller the number of parameters and the amount of computation in the model, the lower the hardware resource consumption. ShuffleNetV2 0.5× used the least amount of resources, but its classification performance was the lowest among light-weight models. On the other hand, MobileNetV3 showed the best classification performance while using fewer hardware resources. Figure 9 shows the comparison result in terms of speed. The left side shows the F1 score and inference speed for each model on the GPU, and the right side shows the results performed on the CPU. Again, MobileNetV3 had the best performance and was processed in the fastest time. MobileNetV2 processed faster than ShuffleNetV2 on the GPU, but was very slow on the CPU. The processing speed in the CPU was in the order of MobileNetV3, CNN-WDI, ShuffleNetV2, ResNet18, and MobileNetV2. As a result of the experiment comparing resource utilization and training and inference speed, the MobileNet V3 model showed the most efficient and good performance in the wafer map classification task.

### 4.5. Ablation Study

In order to find out the effect of the size of the input image, we compared the performance of each model. Table 13 shows the F1 score, number of operations, and throughput when the size of the input image is set to 224 × 224 and 96 × 96 in ShuffleNetV2 and MobileNetV3 with other conditions the same. When the image size was reduced, the amount of computation and throughput improved a lot, but as a result of classification, the F1 score fell significantly to 0.058 for ShuffleNetV2 and 0.048 for MobileNetV3. It was found that the larger the size of the image, the higher the amount of computation and the longer the training time, but the better the classification performance. Depending on the application to be applied, it was necessary to adjust the size of the input image to an appropriate level.

## 5. Conclusions

In this study, we proposed a method that can classify semiconductor wafer map defect patterns using minimal computing resources in a limited hardware environment. We used light-weight convolutional neural network models EfficientNetV2, ShuffleNetV2, MobileNetV2 and V3 for wafer map defect pattern classification and compared them in terms of classification performance, hardware resource usage and execution time. It was shown that the MobileNetV3 could perform with the best classification performance and efficient resource utilization. As a result, by automating the classification of defect map patterns without introducing new hardware in an actual workplace, the manpower required for the task can be reduced, and this will help with the early detection, analysis and action of defects. Currently, at least nine people are undertaking visual inspection of wafer maps for defects in three shifts. If the annual salary of one skilled worker is 50 million won, it has the effect of reducing labor costs by 450 million won a year, and this manpower can be put into other tasks. In addition, by finding a specific pattern through wafer bin map classification, it has the effect of reducing accident handling costs from as little as hundreds of millions of won to as many as billions of won per case. In particular, if an automated wafer map pattern classification system is built using the light-weight model proposed in this study, the cost of purchasing and maintaining high-performance hardware can be saved. Furthermore, it is expected that it can be used as a way to drive deep learning models within limited resources when building systems that deal with similar problems.

## 6. Future Work

The method proposed in this study has a limitation in that it is difficult to recognize the occurrence of a new type of defect pattern. In addition, even when two or more defect patterns are mixed, there is a problem of predicting only one of them. Therefore, future research will be directed in the following two directions. The first is to define one of the defect types as an unknown type, exclude it when training the model, and model it so that it can be recognized as a new type. The second is to train by increasing the number of classes after additionally generating data that mix two or more types in the dataset and adding labels. That way, even images of mixed types will be classified.

## Figures and Tables

**Figure 1 sensors-23-01926-f001:**
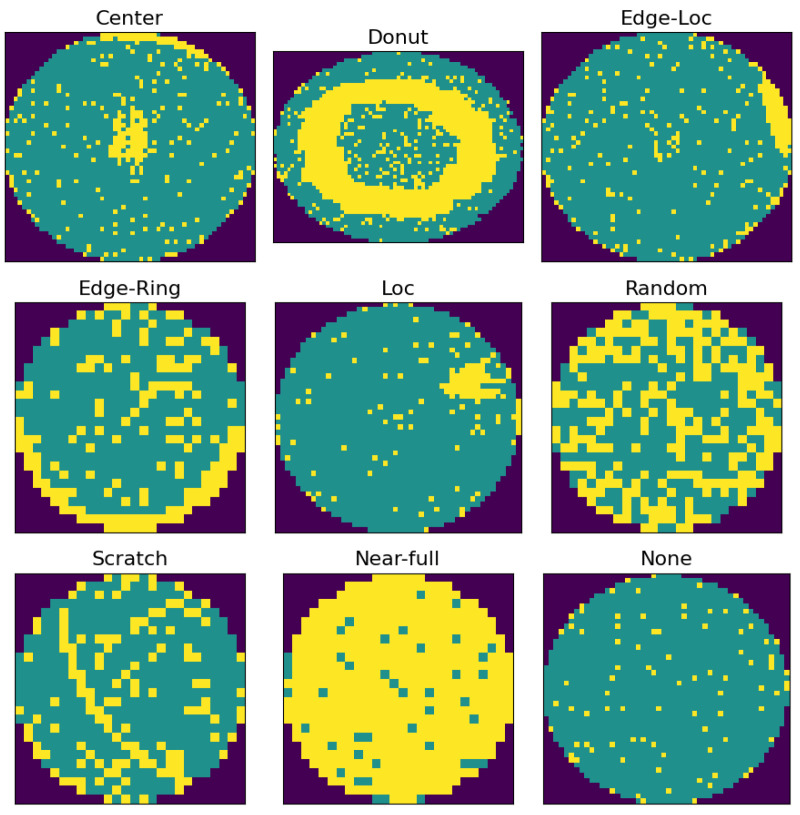
Wafer bin map defect patterns. Green areas are normal chips and yellow areas are defective chips.

**Figure 2 sensors-23-01926-f002:**
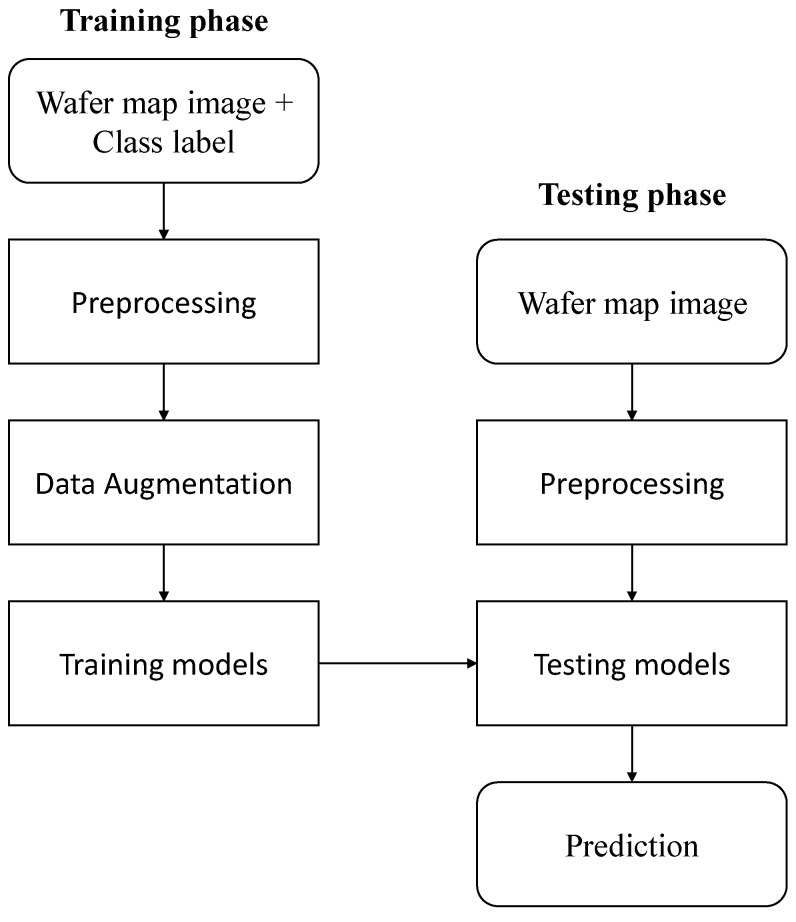
Wafer map classification flowchart.

**Figure 3 sensors-23-01926-f003:**
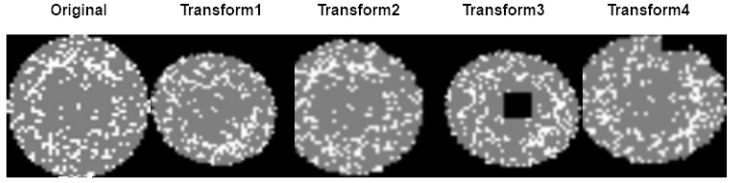
Data augmentation sample.

**Figure 4 sensors-23-01926-f004:**
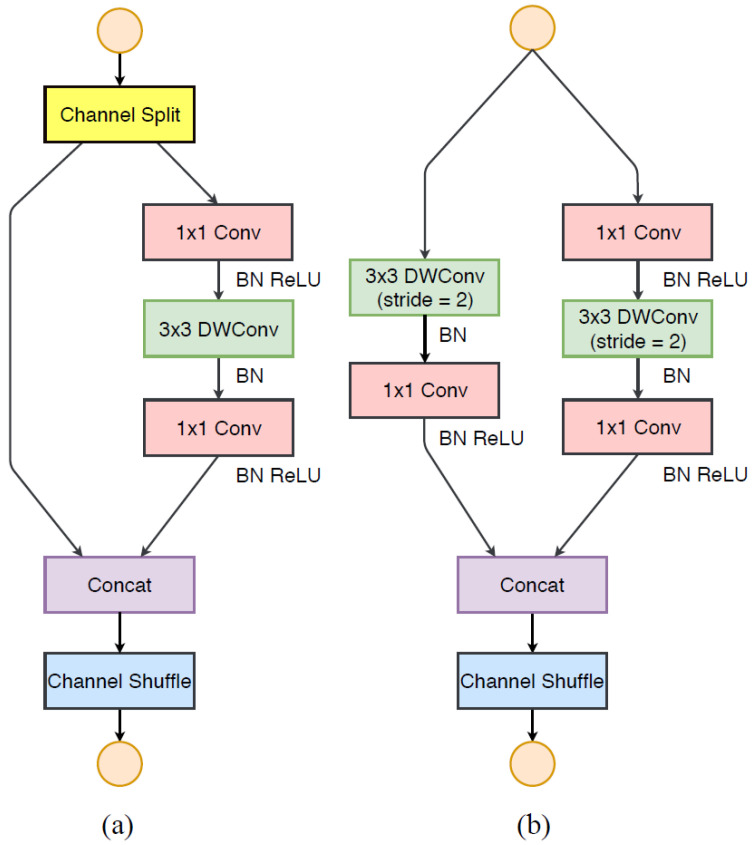
ShuffleNetV2 [22]. (**a**) Basic unit (**b**) A unit for downsampling (2×).

**Figure 5 sensors-23-01926-f005:**
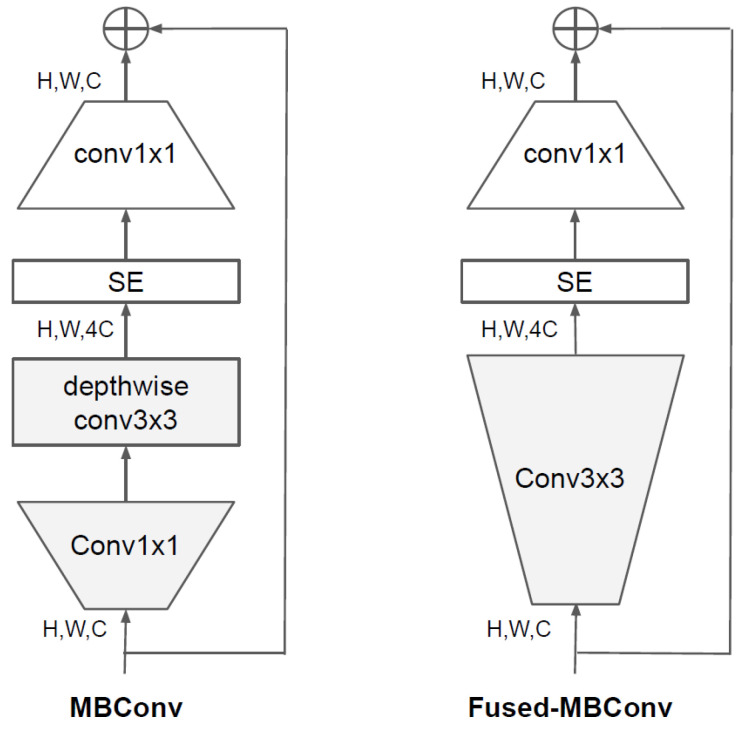
Structure of MBConv and Fused-MBConv [23].

**Figure 6 sensors-23-01926-f006:**
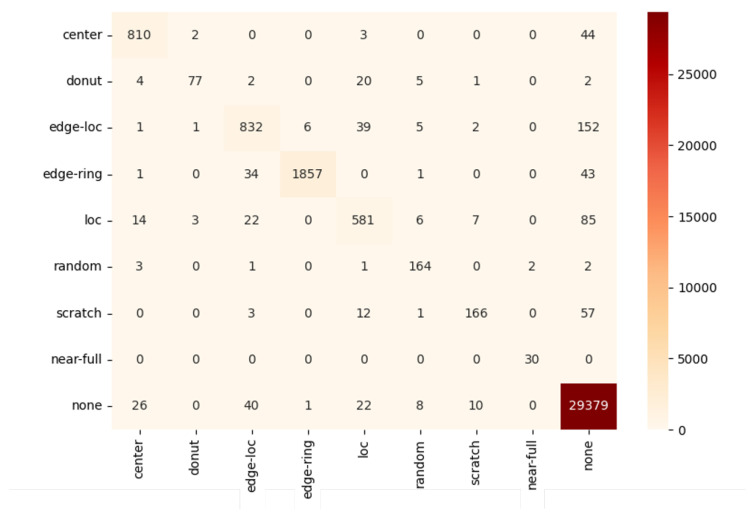
A confusion matrix of MobileNetV3.

**Figure 7 sensors-23-01926-f007:**
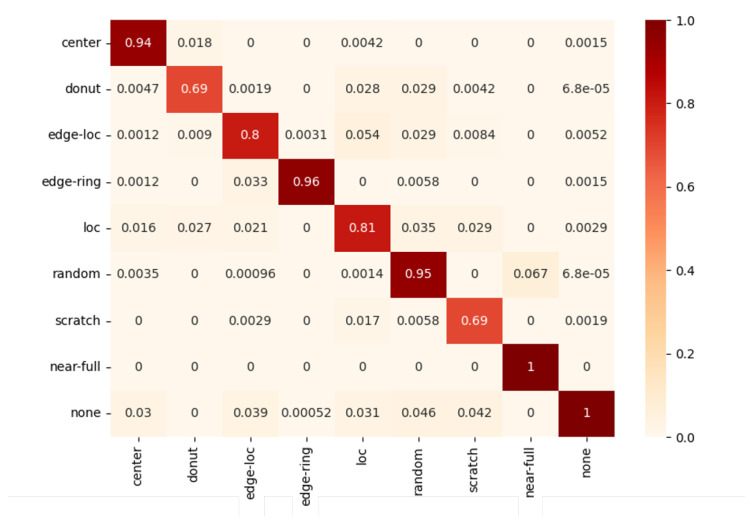
A normalized confusion matrix of MobileNetV3.

**Figure 8 sensors-23-01926-f008:**
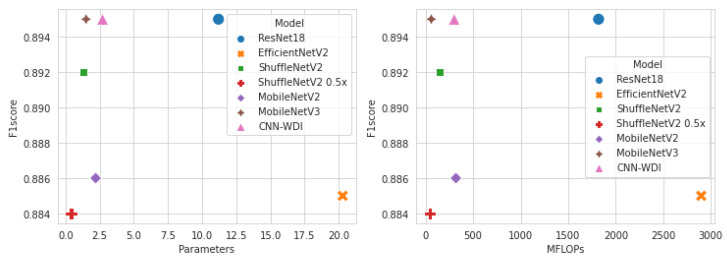
Number of parameters and F1 score (**left**); Number of floating point operations and F1 scores (**right**).

**Figure 9 sensors-23-01926-f009:**
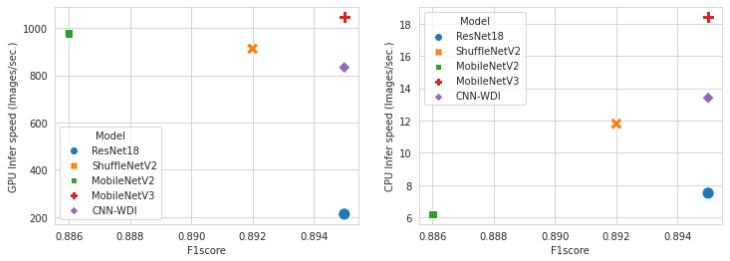
Comparison of F1 score and inference speed; GPU (**left**) and CPU (**right**).

**Table 1 sensors-23-01926-t001:** Classification performance of the WM-811K dataset presented in related works.

Model	Accuracy	F1 Score
CNN-WDI [5]	0.962	-
DMC1 [9]	0.940	-
DCNN+ECOC-SVM [10]	0.964	-
DCNN [11]	0.938	0.938
G2LGAN+MobileNetV2 [12]	0.984	0.930
WM-PeleeNet [14]	0.954	-
ShuffleNet-v2-CNN [15]	0.969	0.957

**Table 2 sensors-23-01926-t002:** Distribution of class labels in the WM-811K dataset.

Class Label	No. of Samples	Proportion
Center	4294	2.48%
Donut	555	0.32%
Edge-loc	5189	3.00%
Edge-ring	9680	5.60%
Loc	3593	2.08%
Near-full	149	0.09%
Random	866	0.50%
Scratch	1193	0.69%
None	147,431	85.24%
Total	172,950	100.00%

**Table 3 sensors-23-01926-t003:** Structure of ShuffleNetV2.

Layer	Output Size	Filter Size	Stride	Repeat	Output Channel	
					0.5×	1×
Input	224 × 224				3	3
Conv1 MaxPool	112 × 112 56 × 56	3 × 3 3 × 3	2 2	1	24	24
Stage2	28 × 28 28 × 28		2 1	1 3	48	116
Stage3	14 × 14 14 × 14		2 1	1 7	96	232
Stage4	7 × 7 7 × 7		2 1	1 3	192	464
Conv5	7 × 7	1 × 1	1	1	1024	1024
Global Pool	1 × 1	7 × 7				
FC					9	9

**Table 4 sensors-23-01926-t004:** Structure of MobileNetV2.

Layer	Input Size	Kernel Size	Stride	Repeat	Output Channel
Conv2d	224 × 224 × 3	3 × 3	2		32
Block1	112 × 112 × 32	3 × 3	1	1	16
Block2	112 × 112 × 15	3 × 3	2	2	24
Block3	56 × 56 × 24	3 × 3	2	3	32
Block4	28 × 28 × 32	3 × 3	2	4	64
Block5	14 × 14 × 64	3 × 3	1	3	96
Block6	14 × 14 × 96	3 × 3	2	3	160
Block7	7 × 7 × 160	3 × 3	1	1	320
Conv2d	7 × 7 × 320	1 × 1	1	1	1280
AvgPool	7 × 7 × 1280	7 × 7		1	
Conv2d	1 × 1 × 1280	1 × 1			9

**Table 5 sensors-23-01926-t005:** Structure of MobileNetV3-Small.

Layer	Input Size	Filter Size	Stride	Output Channel	SE	NL
Conv2d	224 × 224 × 3	3 × 3	2	16		HS
Bneck	112 × 112 × 16	3 × 3	2	16	✓	RE
Bneck	56 × 56 × 16	3 × 3	2	24		RE
Bneck	28 × 28 × 24	3 × 3	1	24		RE
Bneck	28 × 28 × 24	5 × 5	2	40	✓	HS
Bneck	14 × 14 × 40	5 × 5	1	40	✓	HS
Bneck	14 × 14 × 40	5 × 5	1	40	✓	HS
Bneck	14 × 14 × 40	5 × 5	1	48	✓	HS
Bneck	14 × 14 × 48	5 × 5	1	48	✓	HS
Bneck	14 × 14 × 48	5 × 5	2	96	✓	HS
Bneck	7 × 7 × 96	5 × 5	1	96	✓	HS
Bneck	7 × 7 × 96	5 × 5	1	96	✓	HS
Conv2d	7 × 7 × 96	1 × 1	1	576	✓	HS
Pool	7 × 7 × 576	7 × 7	1			
Conv2d, NBN	1 × 1 × 576		1	1024		HS
Conv2d, NBN	1 × 1 × 1024	1 × 1	1	9		

**Table 6 sensors-23-01926-t006:** Structure of EfficientNet V2-S.

Layer	SE	Filter Size	Stride	Repeat	Output Channel
Conv2d		3 × 3	2	1	24
Fused-MBConv1		3 × 3	1	2	24
Fused-MBConv4		3 × 3	2	4	48
Fused-MBConv4		3 × 3	2	4	64
MBConv4	✓	3 × 3	2	6	128
MBConv6	✓	3 × 3	1	9	160
MBConv6	✓	3 × 3	2	15	256
Conv2d		1 × 1		1	1280
Pool		7 × 7		1	
FC				1	9

**Table 7 sensors-23-01926-t007:** The number of training and testing data.

Class Label	Original Training Data	Augmented Training Data	Testing Data
Center	3435	10,000	859
Donut	444	10,000	111
Edge-loc	4151	10,000	1038
Edge-ring	7744	10,000	1936
Loc	2874	10,000	719
Near-full	119	10,000	30
Random	693	10,000	173
Scratch	954	10,000	239
None	117,945	117,945	29,486
Total	138,360	197,945	34,590

**Table 8 sensors-23-01926-t008:** Confusion matrix.

		Prediction	
		Positive	Negative
Actual	Positive	TP; True Positive	FN; False Negative
	Negative	FP; False Positive	TN; True Negative

**Table 9 sensors-23-01926-t009:** Performance comparison of training and validation datasets by model (average ± standard deviation).

Model	Training		Validation	
	Accuracy	F1 Score	Accuracy	F1 Score
ResNet18	$0.986_{\pm 0.005}$	$0.977_{\pm 0.009}$	$0.979_{\pm 0.006}$	$0.966_{\pm 0.011}$
EfficientNetV2	$0.979_{\pm 0.005}$	$0.965_{\pm 0.012}$	$0.973_{\pm 0.005}$	$0.958_{\pm 0.013}$
ShuffleNetV2	$0.978_{\pm 0.006}$	$0.963_{\pm 0.011}$	$0.972_{\pm 0.005}$	$0.952_{\pm 0.012}$
ShuffleNetV2 0.5×	$0.967_{\pm 0.009}$	$0.940_{\pm 0.019}$	$0.962_{\pm 0.008}$	$0.932_{\pm 0.018}$
MobileNetV2	$0.976_{\pm 0.004}$	$0.960_{\pm 0.008}$	$0.972_{\pm 0.004}$	$0.952_{\pm 0.009}$
MobileNetV3	$0.981_{\pm 0.005}$	$0.970_{\pm 0.009}$	$0.975_{\pm 0.005}$	$0.959_{\pm 0.010}$
CNN-WDI [5]	$0.977_{\pm 0.005}$	$0.961_{\pm 0.010}$	$0.970_{\pm 0.007}$	$0.947_{\pm 0.015}$

**Table 10 sensors-23-01926-t010:** Comparison of performance of evaluation datasets by model.

Model	Accuracy	Precision	Recall	F1 Score
ResNet18	0.980	0.915	0.877	0.895
EfficientNetV2	0.979	0.918	0.868	0.885
ShuffleNetV2	0.980	0.929	0.871	0.892
ShuffleNetV2 0.5×	0.979	0.920	0.858	0.884
MobileNetV2	0.979	0.873	0.907	0.886
MobileNetV3	0.980	0.909	0.885	0.895
CNN-WDI [5]	0.979	0.938	0.861	0.895

**Table 11 sensors-23-01926-t011:** Comparison of the number of parameters, memory usage, and computation between models.

Model	Params (M)	Memory (MB)	MAdds	MFLOPs
ResNet18	11.2	22.1	3640	1820
EfficientNetV2	20.3	139	5780	2900
ShuffleNetV2	1.3	18.6	295	149
ShuffleNetV2 0.5×	0.4	8.9	82	42
MobileNetV2	2.2	74.2	625	319
MobileNetV3	1.5	16.1	117	60
CNN-WDI [5]	2.7	8.5	597	301

**Table 12 sensors-23-01926-t012:** Comparison of throughput per unit time between models.

Model	Training (Images/s)		Inference (Images/s)	
	GPU	CPU	GPU	CPU
ResNet18	66.8	2.7	212.4	7.5
EfficientNetV2	171.7	-	544.7	-
ShuffleNetV2	463.8	3.9	912.4	11.8
ShuffleNetV2 0.5×	511.2	-	1026.7	-
MobileNetV2	363.1	2.0	977.5	6.2
MobileNetV3	480.1	5.4	1046.0	18.4
CNN-WDI [5]	416.7	4.7	833.3	13.4

**Table 13 sensors-23-01926-t013:** Performance comparison by image size.

Model	Image Size	F1 Score	MFLOPs	Throughput
ShuffleNetV2	224	0.890	149	463.8
ShuffleNetV2	96	0.832	27	788.5 **(1.7×)**
MobileNetV3	224	0.893	60	480.1
MobileNetV3	96	0.845	12	672.1 **(1.4×)**

## Data Availability

Not applicable.

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
