# Peer review of "Efficient Convolutional Neural Networks for Semiconductor Wafer Bin Map Classification"

_sensors, 2023, doi:10.3390/s23041926_

Round 1

Reviewer 1 Report

Authors have worked to classify Semiconductor Wafer Bin Map using four CNN models.

Abstract should be ended with crisp details. Authors can write results rather than writing just MobileNetV3 is the best performing model.

Authors’ contribution needs to write separately.

Authors are suggested to write organization of paper at end of introduction.

In related work, authors have reviewed less work. It is suggested to add from recent work i.e. from 2021 onwards.

Resolution of figure 3 and 4 need to improve.

It seems that authors have used transfer learning with four pre-trained models. Authors have explained four models in general way. How authors have used these models? – information needs to provide. There are different ways to fine-tune the pre-trained models. How they fine-tuned and are there any FC layers added? It should be explained in methodology section.

Results are derived and presented well.

Comparison with SOTA is recommended.

Conclusion needs to write separately.

Reviewer 2 Report

1. Please be more specific when summarizing the results in Abstract.

2. Technical contributions and novelty of current work should be elaborated in Section 1.

3. Inadequate literature survey in Section 2.1 and more related works should be covered.

4. Authors should also summarize the existing wafer map classification methods in table form in terms of datasets used, classifier used, performance metrics used and any other relevant information for more critical literature analysis.

5. A block diagram should be presented in Section 3 to illustrate the overall idea of proposed work.

6. Figure 2 - Please indicate the types of transformation performed in images denoted as Transform1, Transform2, Transform3 and Transform4.

7. Figure 3 - Why the figure captions starts with (c) and (d)?

8. Is transfer learning used to train the selected networks for wafer defect classification? If yes, authors should mention this mechanisms in the training process.

9. It is more appropriate to rename Section 5 as Conclusion instead of Discussion. 

10. The limitations of current works should be highlighted and appropriate future works should be extended from these limitations. 

Round 2

Reviewer 2 Report

Authors have addressed most of my comments. Just a minor comment for Section 1.3. It will be better to write in paragraph form instead of the current presentation style.

Author Response

We addressed the comments mentioned in the review report. Section 1.3 has been revised to paragraph form.